# APrompt: Attention Prompt Tuning for Efficient Adaptation of Pre-trained Language Models

Qifan Wang[1], Yuning Mao[1], Jingang Wang[2]*, Hanchao Yu[1], Shaoliang Nie[1],
Sinong Wang[1], Fuli Feng[3], Lifu Huang[4], Xiaojun Quan[5], Zenglin Xu[6] and Dongfang Liu[7]*

[1]Meta AI  [2]Meituan Lab  [3]University of Science and Technology of China
[4]Virginia Tech  [5]Sun Yat-sen University  [6]Peng Chen Lab  [7]Rochester Institute of Technology
wqfcr@fb.com, yuningm@fb.com

## Abstract

With the continuous growth of large language models, the process of fine-tuning these models for new tasks has become increasingly parameter-intensive. Prompt tuning, a method that involves tuning a small set of soft prompts, has emerged as an effective and efficient approach for adapting large pre-trained language models. However, most existing prompt tuning approaches only introduce prompts at the input layer, limiting their performance and leaving large rooms for improvement. In this work, we propose a novel Attention Prompt tuning method, namely APROMPT, for efficient adaptation of pre-trained language models. We first demonstrate that existing prompt tuning can be considered as a special case of attention prompt tuning. We then formally introduce APROMPT, which incorporates query, key, and value prompts into the attention layer to guide the attention computation during fine-tuning. Experimental results on the SuperGLUE benchmark consistently demonstrate that our proposed approach outperforms state-of-the-art baselines and full fine-tuning method with pre-trained models at different scales. In addition, a comprehensive set of ablation studies validate the effectiveness of the prompt design, as well as the efficiency of our approach.

## 1 Introduction

Pre-trained Language Models (PLMs) have gained significant popularity in various natural language understanding tasks (Devlin et al., 2019; Lewis et al., 2020; Raffel et al., 2020), exhibiting remarkable success under the *pretrain-then-finetune* paradigm. It has been consistently demonstrated in recent studies (Aribandi et al., 2022; Zhang et al., 2022) that scaling up the size of these models leads to improved performance. Consequently, large language models such as LLaMA 65B (Touvron et al., 2023), GPT-3 175B (Brown et al., 2020),

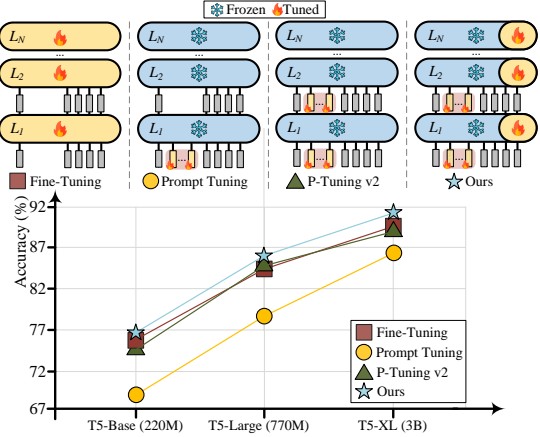

Figure 1: Illustration of APROMPT (ours) and previous works, including Fine-Tuning (Aribandi et al., 2022), Prompt Tuning (Lester et al., 2021), and P-Tuning V2 (Liu et al., 2022) methods. Our method consistently improves over prompt tuning methods and also outperforms fine-tuning method across tasks and model scales.

and PaLM 540B (Chowdhery et al., 2022) are becoming increasingly prevalent. Despite their compelling performance, fine-tuning large-scale PLMs is highly parameter-inefficient due to storing gradients and updating for all model parameters. This inefficiency also arises from the requirement to store and deploy a complete copy of the fine-tuned model for each individual task, resulting in computational expenses that hinder fast model adaptation.

To tackle the challenges associated with full fine-tuning, researchers have proposed parameter-efficient tuning approaches (Guo et al., 2021; He et al., 2022a; Hu et al., 2022) involving techniques such as *partial tuning* and *extra module*. *Partial tuning* methods (Yosinski et al., 2014) focus on fine-tuning only a portion of the backbone, such as the classifier head or the last few layers, while keeping the remaining layers frozen. On the other hand, *extra module* methods introduce learnable bias terms (Cai et al., 2020) or additional adapters (Houlsby et al., 2019) to the network for adaptation. These strategies operate within the *pretrain-*

---
*Corresponding authors.

*then-finetune* paradigm and effectively reduce the number of learnable parameters. However, in general, these approaches tend to underperform the full fine-tuning models with large performance gaps.

Recently, prompt tuning approaches (Lester et al., 2021; Li and Liang, 2021; He et al., 2022b; Yang et al., 2023) have been proposed, which utilize a set of *learnable soft prompts* prepended to the input. These soft prompts consist of continuous embeddings that are updated during the tuning process while keeping the backbone frozen. Prompt tuning offers a conceptually simpler and more flexible method compared to other parameter-efficient tuning approaches. It has been demonstrated to perform closer to full model tuning, especially with large-scale PLMs (Ma et al., 2022; Razdaibiedina et al., 2023a). Prompt tuning provides a promising parameter-efficient alternative to fine-tuning, as the soft prompts used in this approach are typically orders of magnitude smaller, constituting less than 0.5% of the total model parameters. However, most existing prompt tuning approaches have mainly focused on modifying the input layers and have not thoroughly explored the core architecture of the Transformer's self-attention mechanism. Therefore, they often underperform to full fine-tuning and leave substantial room for improvement.

In this paper, we present a novel Attention Prompt tuning approach, APROMPT, for efficient and effective large language model adaptation. We begin by reexamining the prompt tuning approach and establish, both theoretically and empirically, that its input prompts can be considered as specialized key-value prompts. We then formally introduce our APROMPT. Unlike previous prompt tuning methods, APROMPT incorporates three sets of learnable prompts: query, key, and value prompts. These prompts are prepended to the respective matrices in the self-attention block within the Transformer layer. During model tuning, these attention prompts are learned alongside the original input prompts, resulting in more effective guidance of attention computation for new tasks. Evaluation on the SuperGLUE benchmark showcases the superior performance of APROMPT compared to state-of-the-art methods. The ablation study results provide strong evidence for the effectiveness and efficiency of the proposed attention prompts. We summarize the main contributions as follows:

- We establish a connection between existing prompt tuning methods and our approach, demonstrating that input prompts can be viewed as a specialized form of attention prompts. This insight serves as valuable knowledge, enhancing our understanding of both the existing prompt tuning techniques and the novelty of our proposed approach.

- We design novel attention prompt tuning by incorporating query, key, and value prompts into the self-attention computation along with the input prompts. By doing so, these attention prompts play a crucial role in effectively guiding the model's fine-tuning process, enabling faster and more accurate adjustments during the adaptation process.

- We conduct comprehensive experiments on various tasks in the SuperGLUE benchmark, demonstrating the effectiveness of the proposed approach over several state-of-the-art prompt tuning and full fine-tuning methods.

## 2 Related Work

**Pre-trained Language Models** Pre-trained Language Models (PLMs) (Yang et al., 2019; Ainslie et al., 2020; Zaheer et al., 2020; Zhao et al., 2023) have demonstrated huge success across various natural language processing tasks. Pioneering works such as BERT (Devlin et al., 2019) and RoBERTa (Liu et al., 2019) learn contextual representations with masked language model (MLM) and next sentence prediction tasks. Recently, a range of large-scale PLMs, including GPT-3 (Brown et al., 2020), T5 (Raffel et al., 2020), and PaLM (Chowdhery et al., 2022), have emerged with diverse pre-training designs. However, the exponential increase in the number of parameters poses challenges in fine-tuning these models. It becomes computationally expensive to store and maintain all fine-tuned parameters for each tasks.

**Parameter-Efficient Tuning** As the size of PLMs becomes larger, it is increasingly unaffordable to update and save full model copies for each downstream application. Parameter-efficient tuning methods (Pfeiffer et al., 2020, 2021) arise in the era of LLM. Depending on whether new parameters are introduced, we divide parameter-efficient tuning methods into categories of *partial tuning* and *extra module*. Partial tuning methods simply update parts of the model such as the bias term (Zaken et al., 2022) or the last layers (Lee et al., 2019). Extra

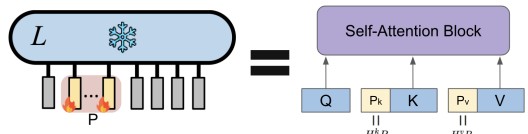

Figure 2: Input prompts $P$ are specialized key and value prompts with $P_k = H^k P$ and $P_v = H^v P$.

module methods introduce task-specific parameters to various locations of the model including Side-Tuning (Zhang et al., 2020) and Adaptors (Hu et al., 2022; Houlsby et al., 2019). There has also been effort unifying different parameter-efficient tuning methods for increased robustness and performance (Mao et al., 2022; He et al., 2022a).

**Prompt Tuning** Prompt tuning (Han et al., 2023; Yan et al., 2023) inserts learnable parameters to the model as virtual tokens where the insertion can happen at the model input (Lester et al., 2021) or each layer (Li and Liang, 2021). Later variants improve prompt tuning methods for NLU (Liu et al., 2022) and NLG (An et al., 2022) tasks respectively through a series of optimization and adaptations. More recent studies add residual connections to improve the performance and stability of prompt tuning (Razdaibiedina et al., 2023b) and extend prompt tuning to the continual learning setting (Razdaibiedina et al., 2023a). However, most of these methods only simply add prompts to input layers, which greatly limited their performances. Most recently, mixture prompt tuning has been proposed in MixPrompt (Yang et al., 2023) and $E^2$VPT (Han et al., 2023), which combines the input prompts with key-value prompts. These methods can be treated as special cases of our attention prompt tuning approach.

## 3 Prompt Tuning Revisit

### 3.1 Preliminary

Prompt tuning methods (Lester et al., 2021; Liu et al., 2022) are proposed as a group of parameter efficient models for fast adaptation of large-scale PLMs to downstream tasks. They introduce a set of task-specific prompts or prompt tokens $P \in R^{d \times m}$, and prepend them to the input sequence $X \in R^{d \times n}$ to form a new input $X_{new} = [P, X] \in R^{d \times (m+n)}$, as shown in left Figure 2. Here $m$ is the length of prompt tokens, $n$ is the input sequence length, and $d$ is the dimension of the embedding vector. These prompts are learned on the downstream task during fine-tuning with the backbone frozen. Prompt tuning achieves promis-

| T5-Large | WSC | CB | Boolq |
|---|---|---|---|
| Prompt Tuning | 78.31 | 88.41 | 83.65 |
| Fixed Key-value Prompts | 78.31 | 88.41 | 83.65 |
| Key-value Prompts | 78.88 | 88.93 | 84.06 |
| Prompt Tuning + Key-value Prompts | **79.12** | **89.17** | **84.25** |

Table 1: Performance (Accuracy) of Prompt Tuning and different key-value prompts variants on WSC, CB and Boolq tasks from SuperGLUE with T5-Large model.

ing results compared to other parameter-efficient tuning methods.

### 3.2 Connection with Key-Value Prompts

In this section, we investigate deeper on how the prompt tuning works and show that *traditional input prompts are equavalent to constrained key-value prompts*. Recall that in prompt tuning, the prompt tokens are first prepended to the input tokens. The new sequence $X_{new} = [P, X]$ is then fed into the Transformer encoder layer to compute the contextual embeddings of the text tokens for the next layer. The self-attention is defined as:

$$Attn([P, X]) = softmax(\frac{Q^T K_{new}}{\sqrt{d}})V_{new}$$

$$Q = H^q X, \ K_{new} = H^k X_{new}, \ V_{new} = H^v X_{new}$$

where $Q$, $K_{new}$ and $V_{new}$ are the new query, key and value embedding matrices, with $H^q$, $H^k$ and $H^v$ as the pre-trained model parameters that are frozen. It is worth noting that for the query $Q$, there is no need to compute a new one since only the original text tokens $X$ are updated and used in the next layer. Then we have:

$$K_{new} = H^k X_{new} = [H^k P, H^k X] = [P_k, K]$$

Similarly, we have $V_{new} = [P_v, V]$. Therefore, we can conclude that adding the prompt tokens $P$ during prompt tuning is equivalent of prepending key prompts $P_k$ and value prompts $P_v$ to the original key and value matrices respectively, as shown in Figure 2. Note that these key-value prompts are constrained or coupled by the input prompts $P$.

### 3.3 Empirical Study

To further validate the findings, we conduct an experiment by comparing four methods, Prompt Tuning (Lester et al., 2021), Fixed Key-value Prompt, Key-value Prompt, and Prompt Tuning + Key-value Prompt, on three tasks from SuperGLUE with T5-Large backone. Fixed Key-value Prompt directly adds fixed key and value prompts computed from

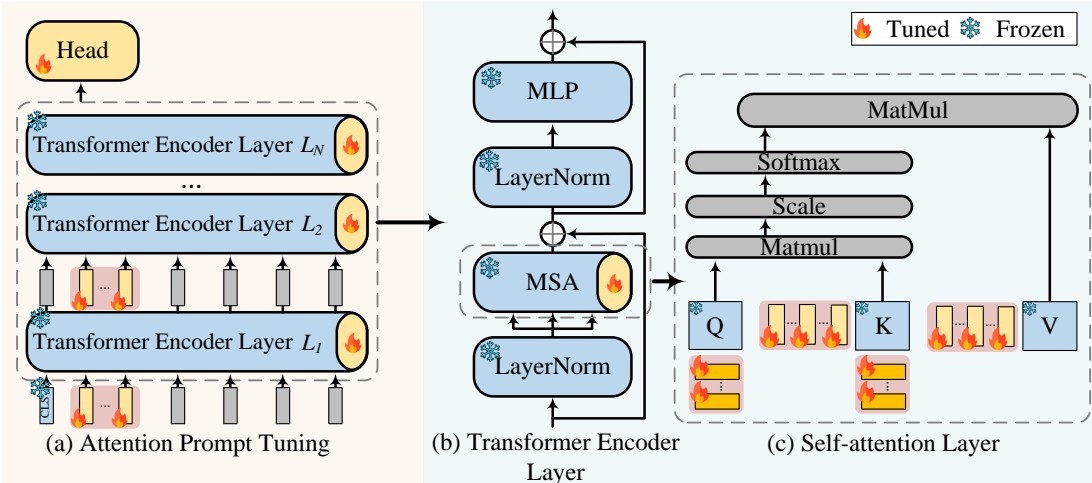

Figure 3: Overview of our APROMPT model. We introduce three sets of trainable attention prompts, namely query, key, and value prompts, during the fine-tuning of the model, in addition to the input prompts. These prompts are incorporated into the query, key, and value matrices within the multi-head attention computation.

the optimal prompts $P^*$ learned in Prompt Tuning, i.e., $P_k = H^k P^*$ and $P_v = H^v P^*$, without any tuning. Key-value Prompt learns the optimal key and value prompts during fine-tuning. Prompt Tuning + Key-value Prompt learns both the input prompts and key-value prompts during fine-tuning.

The comparative results of these methods are presented in Table 1. Firstly, it is clear that the Fixed Key-value Prompts method achieves identical results to Prompt Tuning, which aligns with our expectations and *validates the equivalence between input prompts and constrained key-value prompts*. Secondly, when allowing the key-value prompts to be learned during fine-tuning, we observe improved performance compared to fixed key-value prompts. The reason is that the fixed key-value prompts can be seen as a special case within the search space of unconstrained key-value prompts. Lastly, combining both input prompts and key-value prompts during fine-tuning leads to the highest performance. Our hypothesis is that while key-value prompts theoretically have the potential to encompass the information contained in input prompts, they exist in different embedding spaces. In practice, input prompts can provide additional value during the fine-tuning process. Further analysis and discussion are provided in the experimental section.

## 4 Attention Prompt Tuning

Drawing inspiration from the insights presented in Section 3, we propose an attention prompt tuning approach by introducing the attention prompts in the Transformer layers to facilitate attention

computation. The overall model architecture of APROMPT is depicted in Figure 3. Fundamentally, our model enables the training of only three components while keeping all other parameters frozen. These components are as follows: (1) Input prompts, denoted as $P_i$, which are inserted at the beginning of the input sequence for each Transformer encoder layer. (2) Attention prompts, represented by $P_q$, $P_k$, and $P_v$, are incorporated into the query, key, and value matrices within the self-attention module, respectively. These prompts allow the model to learn new attention patterns from the fine-tuning data. (3) A task-specific head, which is a lightweight module dedicated to the specific task and can be trained efficiently.

### 4.1 Input Prompts

In a similar vein to traditional prompt tuning methods (Lester et al., 2021; Li and Liang, 2021), input prompts consist of a set of $d$-dimensional embedding vectors, where the dimensionality matches that of the text tokens. These prompts are inserted at the beginning of the input sequence in each Transformer encoder layer and interact with all the text tokens. They facilitate the learning of task-specific embeddings, effectively guiding the model's performance on new tasks.

Formally, these input prompts are defined as $P_i = \{P_i^1, P_i^2, \ldots, P_i^N\}$, where $P_i^j$ denotes the learnable input prompts in the $j_{th}$ Transformer encoder layer, and $N$ is the total number of layers. Then the encoder layers are represented as:

$$Z^1 = L_1(P_i^1, E)$$

$$Z^j = L_j(P_i^j, \ Z^{j-1}) \quad j = 2, 3, \ldots, N$$

where $Z^j$ represents the contextual embeddings of the text tokens computed by the $j_{th}$ encoder layer. The different colors indicate trainable and frozen parameters, respectively. $E$ is the text token embeddings initialized from the backbone.

## 4.2 Attention Prompts

While input prompts are effective in acquiring knowledge about the new task, they do not possess the capability to guide the interaction of information within each encoder layer. During fine-tuning on a new task with new data, the word distribution may differ significantly from the examples seen during pre-training. Consequently, it becomes imperative to enhance the model's capacity for capturing new information from the fine-tuning data. This entails enabling better attention among the input tokens to effectively learn new patterns that emerge in the task-specific context.

Figure 4: The new query, key and value matricies.

In order to address this, we introduce a novel set of attention prompts that are integrated into the attention block within each encoder layer. These attention prompts can be categorized into two groups: *query-key prompts* and *key-value prompts*. The query-key prompts, denoted as $P_q^{QK}$ and $P_k^{QK}$, consist of small matrices (comprising a few rows) that are appended to the original query and key matrices within the attention module. By incorporating these query-key prompts, we enhance the computation of attention maps among the tokens, thereby improving the attention mechanism. The key-value prompts, represented by $P_k^{KV}$ and $P_v^{KV}$, which are two supplementary matrices (a few columns) inserted to the key and value matrices, respectively. These key-value prompts provide additional information for the input tokens to attend to, thereby enhancing the representation of the learned embeddings. By incorporating both query-key prompts and key-value prompts, we aim to enable more effective information interaction and capture new patterns during the fine-tuning process. The new query, key and value matrices are augmented with these new attention prompts as shown in Figure 4. It is worth noting that the new key matrix is appended by both the key prompts from query-key and key-value prompts. Then the new attention computations are:

$$L(\cdot) = \text{MLP} \, ( \, \text{LN} \, (\text{MSA} \, (\cdot) \, ) \, )$$

$$\text{MSA}(\cdot) = softmax(\frac{Q_{new}^T K_{new}}{\sqrt{d}})V_{new}$$

where MLP and LN are the frozen multi-layer perceptron and layer norm, and MSA is the multi-head self-attention inside the Transformer encoder layer. In this way, the attention prompts can effectively guide the model adaptation to the new task.

## 4.3 Task-specific Head

For each downstream task, we also fine-tune a task-specific head, which is a very small module dedicated to the specific task to generate the predictions.

$$y = \text{Head}(Z^N)$$

where $Z^N$ is the output contextual embedding from the top layer of the encoder.

## 5 Experiments

### 5.1 Datasets

Following previous works on prompt tuning (Lester et al., 2021; Liu et al., 2022), we use NLU tasks from the SuperGLUE benchmark to evaluate the performance of the language model (Raffel et al., 2020; Aribandi et al., 2022). Specifically, we use the following 8 datasets: BoolQ (Clark et al., 2019), CB (Jiang and de Marneffe, 2019), COPA (Roemmele et al., 2011), MRC (Khashabi et al., 2018), ReC (Zhang et al., 2018), RTE (Giampiccolo et al., 2007), WiC (Pilehvar and Camacho-Collados, 2019) and WSC (Levesque et al., 2012). More details are provided in the Appendix.

### 5.2 Baselines

Our model is compared with five state-of-the-art prompt tuning and fine-tuning methods.

**Fine-Tuning** (Aribandi et al., 2022) is the standard full fine-tuning approach of T5, where all the pre-trained parameters are fine-tuned.

**Prompt-Tuning** (Lester et al., 2021) is the vanilla prompt tuning approach which adds the input prompts in the first input layer.

**P-Tuning v2** (Liu et al., 2022) builds on top of Prompt-Tuning by inserting a set of individual prompts to each Transformer layers.

**XPrompt** (Ma et al., 2022) designs more efficient prompts by pruning the least important token-level and piece-level prompts.

| Method | Para | Boolq Acc | CB F1/Acc | COPA Acc | MRC F1/EM | ReC F1/EM | RTE Acc | WiC Acc | WSC Acc | Average Score |
|---|---|---|---|---|---|---|---|---|---|---|
| **T5-Base** (220M) | | | | | | | | | | |
| Fine-Tuning* (Aribandi et al., 2022) | 100% | **82.30** | 91.30 | 60.00 | 58.25 | 80.55 | 84.50 | 69.30 | **81.70** | 76.10 |
| Prompt-Tuning (Lester et al., 2021) | 0.06% | 78.12 | 84.42 | 54.37 | 51.14 | 71.35 | 75.27 | 62.29 | 67.36 | 68.04 |
| P-Tuning v2 (Liu et al., 2022) | 0.53% | 80.81 | 90.23 | 61.28 | 55.64 | 78.13 | 81.98 | 67.56 | 78.32 | 74.35 |
| XPrompt (Ma et al., 2022) | 0.04% | 79.67 | 86.72 | 56.95 | 53.08 | 74.36 | 78.29 | 64.31 | 73.68 | 70.88 |
| ResPrompt (Razdaibiedina et al., 2023b) | 0.21% | 79.25 | 85.33 | 58.64 | 52.91 | 73.19 | 77.14 | 62.36 | 70.82 | 69.95 |
| APROMPT (Ours) | 0.45% | 81.83 | **91.86** | **61.54** | **59.07** | **81.18** | **85.76** | **69.50** | 81.49 | **76.84** |
| **T5-Large** (770M) | | | | | | | | | | |
| Fine-Tuning* (Aribandi et al., 2022) | 100% | 88.30 | 95.35 | 87.00 | 67.25 | 87.85 | **90.60** | 73.50 | 88.50 | 84.47 |
| Prompt-Tuning (Lester et al., 2021) | 0.03% | 83.65 | 88.41 | 82.67 | 63.28 | 82.46 | 85.19 | 71.05 | 78.31 | 79.25 |
| P-Tuning v2 (Liu et al., 2022) | 0.48% | 87.92 | 95.56 | 86.20 | 70.47 | 89.03 | 89.14 | 71.81 | 86.59 | 84.59 |
| XPrompt (Ma et al., 2022) | 0.02% | 85.54 | 91.39 | 85.05 | 67.32 | 85.47 | 87.30 | 73.22 | 80.28 | 81.95 |
| ResPrompt (Razdaibiedina et al., 2023b) | 0.15% | 83.51 | 90.64 | 82.79 | 65.16 | 84.72 | 86.97 | 71.13 | 80.36 | 80.66 |
| APROMPT (Ours) | 0.37% | **90.35** | **95.83** | **88.32** | **71.98** | **90.64** | 90.47 | **74.67** | **90.13** | **86.55** |
| **T5-XL** (3B) | | | | | | | | | | |
| Fine-Tuning* (Aribandi et al., 2022) | 100% | 89.60 | 94.20 | **96.00** | 76.15 | 92.05 | 91.70 | 74.30 | 95.20 | 88.65 |
| Prompt-Tuning (Lester et al., 2021) | 0.01% | 87.58 | 91.25 | 91.56 | 73.49 | 90.14 | 89.35 | 74.21 | 87.16 | 85.59 |
| P-Tuning v2 (Liu et al., 2022) | 0.45% | 90.11 | 94.08 | 95.33 | 75.21 | 92.39 | 92.13 | 75.46 | 94.25 | 88.62 |
| XPrompt (Ma et al., 2022) | 0.01% | 89.14 | 92.73 | 95.18 | 75.01 | 91.18 | 92.16 | 74.85 | 89.43 | 87.46 |
| ResPrompt (Razdaibiedina et al., 2023b) | 0.04% | 88.46 | 92.54 | 93.12 | 75.17 | 91.20 | 91.64 | 75.32 | 89.15 | 87.08 |
| APROMPT (Ours) | 0.32% | **90.72** | **95.48** | 95.83 | **78.68** | **93.75** | **93.36** | **76.43** | **96.17** | **90.05** |

Table 2: Performance comparison result (%) on SuperGLUE development set. '*' indicates the results reported in (Aribandi et al., 2022). 'Para' is the number of trainable parameters. The best results are in bold with underline representing the second best ones. For tasks with two metrics, the average score is reported. All scores are averaged over 5 runs. Results are statistically significant with respect to all baselines on each PLM (all p-value < 0.005).

**ResPrompt** (Razdaibiedina et al., 2023b) adds residual connections to improve the performance and stability of prompt tuning.

## 5.3 Implementation Details

APROMPT is implemented with the OpenPrompt library (Ding et al., 2022), which is a unified and extensible toolkit for prompt tuning research. Our model is trained on 16 NVIDIA Tesla V100 GPUs. We translate each SuperGLUE dataset into a text-to-text format following (Raffel et al., 2020). Three scales pre-trained models are used: T5-Base, T5-Large and T5-XL with 200M, 770M and 3B parameters, respectively. Following previous studies (Lester et al., 2021; Ma et al., 2022), we train our prompts for 100 epochs with a constant learning rate of 0.3 and a batch size of 16. There are three hyperparameters in our model: the lengthes of input, query-key and key-value prompts. For our method, we set the number of input prompts to 10 (as we found our model is less sensitive to it), and linearly search the best prompt length for both query-key and key-value prompts from {1, 5, 10, 20, 50}. For all prompt tuning baselines, we search the best input prompt length from {5, 10, 20, 50, 100}. The best checkpoints are selected via early stopping on the development set. The models are trained using the Adafactor (Shazeer and Stern,

2018) optimizer with weight decay $1e^{-5}$.

## 5.4 Main Results

The main performance comparison results are presented in Table 2. There are several key observations from these results. **First**, APROMPT consistently outperforms all prompt tuning baselines across different backbone models, showcasing the effective design of its attention prompts. For instance, when evaluating on Boolq with T5-Large, the Acc score of APROMPT demonstrates a significant improvement of 5.62% and 2.76% compared to two strong methods, XPrompt and P-Tuning v2, respectively. This highlights the limitation of existing prompt tuning approaches that primarily focus on designing input prompt tokens, failing to capture the intricate token interactions within new data. In contrast, the attention prompts employed by our approach successfully bridge this gap, resulting in enhanced performance. **Second**, APROMPT outperforms the full fine-tuning method in most cases, while other prompt tuning baselines still exhibit certain gaps, particularly when using smaller backbone models like T5-Base. This observation highlights the effectiveness of our approach across a range of natural language understanding tasks. Moreover, our model achieves these results while training only around 0.4% of the parame-

| Model | Boolq | CB | RTE | WSC |
|---|---|---|---|---|
| Fine-Tuning | 71.31 | 75.72 | 70.35 | 67.52 |
| Prompt-Tuning | 69.48 | 75.16 | 71.73 | 67.47 |
| P-Tuning v2 | 73.36 | 77.94 | 73.48 | 70.50 |
| XPrompt | 71.53 | 76.57 | 72.68 | 69.75 |
| ResPrompt | 70.38 | 76.20 | 70.85 | 68.35 |
| APROMPT (Ours) | **75.66** | **78.51** | **75.83** | **72.30** |

Table 3: Performance comparision results with few-shot (32 samples) setting on Boolq, CB, RTE and WSC tasks for the T5-XL model. APROMPT consistently outperforms all baselines in low resource scenarios.

ters in the backbone, making it significantly more parameter-efficient than the full fine-tuned model. It is worth noting that although APROMPT introduces additional attention prompts, the length of the input prompts are largely reduced (fixed to 10) and thus resulting in even less total trainable parameters compared with P-Tuning v2. **Third**, it is worth noting that the gap between fine-tuning and other prompt tuning methods diminishes as the size of the backbone models increases. This finding aligns with previous studies (Lester et al., 2021; Ma et al., 2022) and underscores the trend of convergence between fine-tuning and alternative prompt tuning approaches.

# 6 Analysis and Discussion

**Results on Low-resource Scenario** We conducted further evaluations to assess the performance of APROMPT and other baseline models in a low-resource setting. Following (Schick and Schütze, 2021), we randomly selected 32 examples for each task as the new training set, using a fixed random seed. We fine-tuned the prompt model on this limited training set and reported the results on the full dev set using the best checkpoint in Table 3. It is evident that all methods experience a significant drop in performance due to the limited data available for training. Nevertheless, APROMPT consistently outperforms the baseline models on tasks such as Boolq, CB, RTE, and WSC. Additionally, we observe that most prompt tuning approaches achieve better results compared to fine-tuning, indicating that despite the challenges of overfitting when training with limited data, prompt tuning methods exhibit superior generalization capabilities compared to full fine-tuning.

**Impact of Different Prompts** To investigate the impact of different prompts in our model, we conducted an ablation study by exploring various prompt combinations in APROMPT. Specif-

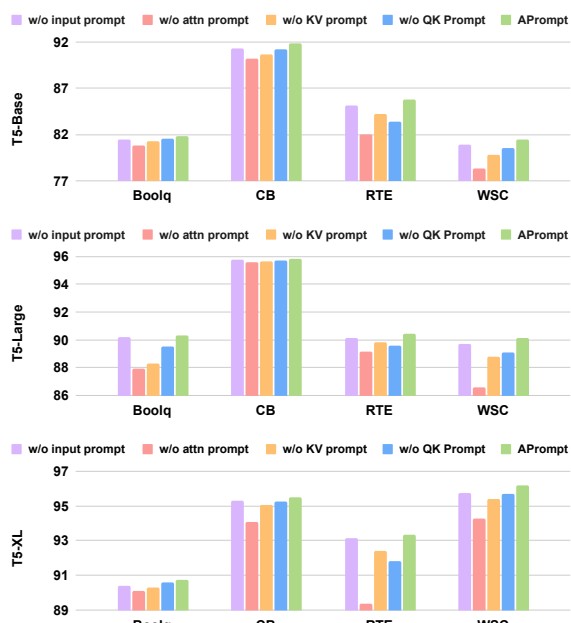

Figure 5: Ablation study on the impact of different prompt combinations on four tasks of SuperGLUE.

ically, we experimented with four additional models: one without input prompts, one without query-key prompts, one without key-value prompts, and one without both query-key and key-value prompts. The Acc scores obtained on four SuperGLUE tasks with different backbone models are presented in Figure 5. The results reveal that the model's performance drops when any of the trainable prompts is removed, which aligns with our expectations. Furthermore, we observed that the performance drop of APROMPT without input prompts is relatively small compared to the models without attention prompts. This suggests the significance of both query-key and key-value prompts in comparison to input prompts, thereby validating the analysis presented in section 3. Once again, it is worth noting that combining all prompts in APROMPT leads to the best performance.



Figure 6: Ablation study of query-key and key-value prompt lengths. We vary the number of prompts for different combinations, and evaluate (Acc) on RTE and WSC tasks with T5-XL as the backbone model.

**Impact of Prompt Length** In APROMPT, the lengths of query-key prompts and key-value

| Position | Boolq | CB | RTE | WSC |
|---|---|---|---|---|
| First-layer | 78.56 | 85.48 | 90.28 | 90.88 |
| First 12-layers | 88.35 | 88.64 | 92.42 | 94.32 |
| Last-layer | 75.82 | 84.37 | 90.75 | 91.15 |
| Last 12-layers | 82.55 | 88.49 | 92.67 | 95.36 |
| Alternative 12-layers | 90.31 | 90.14 | 92.74 | 95.21 |
| APROMPT (All) | **90.72** | **95.48** | **93.36** | **96.17** |

Table 4: Performance comparison with different prompt positions on Boolq, CB, RTE and WSC for T5-XL.

prompts are the only hyperparameters that require tuning. To further analyze the impact of different prompt lengths on model performance, we conducted an ablation study by modifying both prompt lengths across $\{1, 5, 10, 20, 50\}$. We experiment over all possible length combinations, and a detailed discussion on how to balance these two prompts will be provided in later experiments. The model performance results of all prompt length combinations on RTE and WSC are shown in Figure 6. It can be seen that there is no universal optimal prompt length that consistently achieves the best performance across both tasks. For instance, on RTE, the highest score is obtained with 10 key-value prompts and 5 query-key prompts, while on WSC, the best performance is achieved with 20 key-value prompts and 10 query-key prompts. We hypothesize that different tasks and datasets exhibit distinct data distributions, with 'hard' tasks potentially requiring longer prompts to effectively capture the underlying patterns and knowledge within the data. However, this comes with the trade-off of an increased number of trainable parameters. Nonetheless, we observed that our model's performance remains relatively stable within a certain range of prompt lengths.

**Impact of Prompt Positions**  This study evaluates the impact of prompt positions to the model performance. Concretely, we train five additional models with different prompt locations (applied to both encoder and decoder), including only first layer, last layer, first 12 layers, last 12 layers and alternative 12 layers. The performance comparison results on MAVE are reported in Table 4. It is not surprising to see that inserting prompts to all encoder and decoder layers achieves the best performance. We can also observe that only putting the prompts to the first (input) or last (output) layer results in large performance drops, which is consistent with the observations in other prompt tuning works (Liu et al., 2022; Jia et al., 2022).

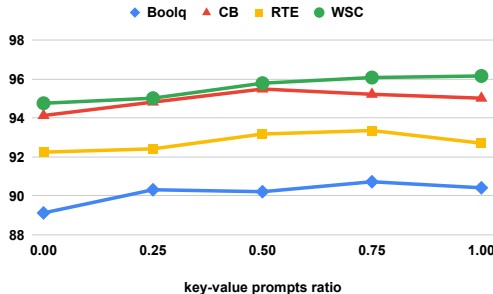

Figure 7: Performance comparison of different ratios of the key-value prompts when fixing the total number of trainable parameters using T5-XL.

**Effect of Attention Prompts Balancing**  The query-key and key-value prompts in APROMPT contribute differently to the model performance. To further investigate their correlation and effectiveness, we conduct an experiment by fixing the total number of trainable parameters, and adjusting the ratio of key-value prompts from $\{0, 0.25, 0.5, 0.75, 1\}$. The model performances at different ratios on four SuperGLUE tasks are illustrated in Figure 7. We observe slightly different patterns on different tasks. For example, on WSC, key-value prompts with 0.75 ratio achieves the best score, while key-value prompts with 0.5 gives the best performance on CB. Nevertheless, APROMPT with ratio 0 (no key-value prompts) or 1 (no query-key prompts) underperforms other prompt combinations in most cases, indicating the effectiveness of both attention prompts.

**Variants of APROMPT**  We compare APROMPT with its two variants to analyze the performance-scale trade-off. Firstly, we remove the input prompts, retaining only the attention prompts, resulting in a variant named APROMPT-. Additionally, we apply the pruning technique in XPrompt (Ma et al., 2022) to eliminate the least important prompts, creating a variant called APROMPT +. From the results in Table 5, we observe that the contribution of input prompts is not particularly significant, aligning with the findings in section 3. When applying prompt trimming, we note that while the number of trainable parameters decreases, the performance behavior varies across different tasks, leaving a room for further exploration in terms of finding the optimal balance between the number of prompts and model performance.

## 7 Conclusions

This work first connects existing prompt tuning with attention prompt tuning and show that input

| Model | Para | Boolq | CB | RTE | WSC |
|---|---|---|---|---|---|
| APROMPT- | 0.25% | 90.41 | 95.28 | 93.12 | 95.82 |
| APROMPT | 0.32% | 90.72 | 95.48 | 93.36 | 96.17 |
| APROMPT + | 0.21% | 91.05 | 95.34 | 93.51 | 96.02 |

Table 5: Different variants of our model with T5-XL.

prompts can be considered as a special case of key-value prompts in the attention layer. Inspired by the observation, we introduce three sets of new prompts, namely query, key, and value prompts, and incorporate them into the attention layer to guide the attention computation during fine-tuning. Experimental results on SuperGLUE demonstrate the superior performance of our model over several state-of-the-art baselines.

## Limitations

There are two limitations of our APROMPT model. First, while APROMPT outperforms other prompt tuning and fine-tuning approaches, identifying the optimal combination of attention prompts automatically poses a challenge that remains unanswered. In our experiments, we conduct grid search to empirically determine the optimal prompt length. However, in the future, we intend to explore a systematic solution that can identify either the optimal combination or a suboptimal one. Second, our current model learns task-specific prompts for each individual task. To address this, we plan to investigate a parametric network that can guide the learning of task-agnostic prompts, thereby enhancing the model's flexibility and adaptability across multiple tasks.

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

# A Dataset Statistics

Table 6 shows details of the eight datasets from SuperGLUE benchmark (Wang et al., 2019) that we used for our experiments, along with their training sizes and evaluation metrics. Following (Raffel et al., 2020) and (Lester et al., 2021), for tasks that have two evaluation metrics we use the average of both scores as the final performance metric.

| Dataset | Examples | Task | Domain | Metric |
|---------|----------|------|--------|--------|
| Boolq | 9,427 | QA | Wikipedia | Acc |
| CB | 250 | NLI | various | F1/Acc |
| COPA | 400 | QA | blogs, encyclop | Acc |
| MRC | 5,100 | QA | various | F1/EM |
| ReC | 101,000 | QA | various | F1/EM |
| RTE | 2,500 | NLI | news, Wiki | Acc |
| WiC | 6,000 | WSD | lexical databases | Acc |
| WSC | 259 | coref. | fiction books | Acc |

Table 6: The details of 8 SuperGLUE tasks used in our experiments. NLI denotes natural language inference, QA denotes questions and answers task, WSD denotes word sense disambiguation, EM denotes exact match scoring, Acc denotes accuracy.

# B Training Details

## B.1 Settings

Our model is implemented with the OpenPrompt library (Ding et al., 2022), which is a unified and extensible toolkit for prompt tuning research. Our model is trained on 16 NVIDIA Tesla V100 GPUs. We translate each SuperGLUE dataset into a text-to-text format following (Raffel et al., 2020). We train our prompts for 100 epochs with a constant learning rate of 0.3 and a batch size of 16. There are three hyperparameters in our model: the lengthes of input, query-key and key-value prompts. For our method, we set the number of input prompts to 10 across all tasks, and linearly search the best prompt length for both query-key and key-value prompts from {1, 5, 10, 20, 50}. For all prompt tuning baselines, we search the best input prompt length from {5, 10, 20, 50, 100}. The best checkpoints are selected via early stopping on the development set. Adafactor (Shazeer and Stern, 2018) optimizer is used in model training with weight decay $1e^{-5}$.

## B.2 Tokenization and Preprocessing

Following common practice (Lester et al., 2021), for all our experiments, we set the maximum input length (including the input prompt) to 512 tokens. We use padding to maximum length and mask out the padded tokens. In case of input exceeding 512 tokens, we truncate the input. We do not perform any specific text preprocessing (e.g. removing punctuation) but instead directly tokenize the raw text from SuperGLUE datasets using the corresponding model tokenizer. For all experiments, we follow T5 (Raffel et al., 2020) formatting. We feed input examples along with their descriptors (e.g. 'sentence1' and 'sentence2'), and cast all classification tasks into text-to-text format (e.g. 0 and 1 classes in Boolq task are cast into 'True' and 'False') replicating guidelines from T5.

## B.3 Prompt initialization

In our experiments, we initialize input prompts using randomly sampled vocabulary embeddings similar to (Lester et al., 2021). We sample uniformly across the whole vocabulary, without limiting to top-k most common tokens. The attention prompts are randomly initialized.

| Methods | Boolq | CB | RTE | WSC |
|---------|-------|------|------|------|
| Fine-Tuning (Aribandi et al., 2022) | 89.60 | 94.20 | 91.70 | 95.20 |
| Partial-1 (Yosinski et al., 2014) | 83.28 | 84.66 | 83.92 | 87.51 |
| Adapter (Pfeiffer et al., 2020) | 85.37 | 86.54 | 85.75 | 89.17 |
| BitFit (Zaken et al., 2022) | 85.91 | 90.53 | 86.34 | 90.48 |
| LoRA (Hu et al., 2022) | 89.48 | 94.66 | 91.91 | 95.82 |
| APROMPT (Ours) | **90.72** | **95.48** | **93.36** | **96.17** |

Table 7: Performance comparison with other non-prompt tuning based parameter efficient methods on Boolq, CB, RTE and WSC for T5-XL.

# C Comparison with Other Parameter Efficient Methods

To conduct a full performance evaluation, we further conduct comparision of our approach with other non-prompt tuning based parameter efficient methods, including Parial tuning (Yosinski et al., 2014) (Partial-1 means only fine-tuning the first layer), Adapter (Pfeiffer et al., 2020), BitFit (Zaken et al., 2022) and LoRA (Hu et al., 2022). The performance comparision results are reported in Table 7. It can be seen that APROMPT outperforms all the parameter efficient methods with large margins. In fact, existing prompt tuning approaches also achieve better performances compared to these baselines.

| Methods | Boolq | CB | COPA | MRC | ReC | RTE | WiC | WSC |
|---------|-------|-----|------|------|--------|-------|-------|-----|
| Prompt-Tuning | 2h38m | 8m | 11m | 46m | 18h25m | 57m | 51m | 8m |
| P-Tuning v2 | 3h36m | 10m | 14m | 1h15m | 20h12m | 1h28m | 1h14m | 12m |
| XPrompt | 4h47m | 14m | 29m | 1h53m | 28h41m | 2h26m | 2h19m | 19m |
| ResPrompt | 3h47m | 10m | 23m | 1h21m | 22h32m | 1h51m | 1h34m | 12m |
| APROMPT | 3h21m | 9m | 15m | 1h7m | 21h | 1h24m | 1h16m | 10m |

Table 8: Training time of APROMPT on SuperGLUE.

## D  Training Time

We further discuss and report the training time of different methods on all the tasks in SuperGLUE in Table 8. For Prompt-Tuning, the trainable parameters consist of the prompts designed for the input layer. In the case of P-Tuning V2, the trainable parameters encompass the prompts associated with all layers. XPrompt focuses on trainable parameters related to the pruned prompts, specifically for the input layer, following the pruning process. As for ResPrompt, the trainable parameters include both the input prompts and the residual network components. The total count of trainable parameters for each approach is detailed in Table 2.