# OpenReview forum: "APrompt: Attention Prompt Tuning for Efficient Adaptation of Pre-trained Language Models"
_EMNLP/2023/Conference — EMNLP 2023 Main_

### Official Review · Reviewer_S6E8 · 2023-08-02

**Soundness:** 4

**Excitement:**

4: Strong: This paper deepens the understanding of some phenomenon or lowers the barriers to an existing research direction.

**Missing References:**

[1] LoRA: Low-Rank Adaptation of Large Language Models.
[2] A Survey of Large Language Models.
[3] Challenges and Applications of Large Language Models

**Paper Topic And Main Contributions:**

This paper proposes an attention prompt tuning method for parameter-efficient tuning PLMs. This paper has proved that existing prompt tuning can be considered as a special case of attention prompt tuning. The main contributions lie in that extensive experiments has verified the effectiveness of this kind of prompt tuning.

**Questions For The Authors:**

1. You should make a comparison among prompt tuning methods including their detailed trainable parameters and training time.
2. LoRA [1] is an important parameter-efficient tuning method for large language models. You should compare with it.
[1] LoRA: Low-Rank Adaptation of Large Language Models

**Reasons To Accept:**

1. Novel method. The proposed attention prompt tuning is novel compared to previous method and can be considered as a unified form.
2. Insightful analysis. The empirical study is useful to build a connection between the proposed tuning and previous tuning.
3.  Extensive experiments. The authors conduct many experiments to validate the proposed method.
4. Good writing.

**Reasons To Reject:**

1. Missing some detailed comparison with previous methods.
2. Missing some important baselines.

**Reproducibility:**

4: Could mostly reproduce the results, but there may be some variation because of sample variance or minor variations in their interpretation of the protocol or method.

**Reviewer Confidence:**

5: Positive that my evaluation is correct. I read the paper very carefully and I am very familiar with related work.

---

> ### Author Rebuttal · Authors · 2023-08-27
>
> We sincerely thank you for your thoughtful feedback! We discuss your raised points as follows:
>
> #### **Q1: Detailed trainable parameters and training time of prompt tuning methods.**
>
> **A1:** Thank you for your excellent suggestion. In the context of these different techniques:
>
> * For Prompt-Tuning, the trainable parameters consist of the prompts designed for the input layer.
> * In the case of P-Tuning V2, the trainable parameters encompass the prompts associated with all layers.
> * XPrompt focuses on trainable parameters related to the pruned prompts, specifically for the input layer, following the pruning process.
> * As for ResPrompt, the trainable parameters include both the input prompts and the residual network components.
>
> The total count of trainable parameters for each approach is detailed in Table 2. Furthermore, we report the training time of all methods below. We’ll supplement the above discussion and the training time results in our revision.
>
> |  Method  | Boolq | CB | COPA | MRC | ReC | RTE | WiC | WSC |
> | :-: | :-: | :-: | :-: | :-: | :-: | :-: | :-: | :-: |
> | Prompt-Tuning | 2h38m | 8m | 11m | 46m | 18h25m | 57m | 51m | 8m |
> | P-Tuning v2 | 3h36m | 10m | 14m | 1h15m | 20h12m | 1h28m | 1h14m | 12m |
> | XPrompt | 4h47m | 14m | 29m | 1h53m | 28h41m | 2h26m | 2h19m | 19m |
> | ResPrompt | 3h47m | 10m | 23m | 1h21m | 22h32m | 1h51m | 1h34m | 12m |
> | APrompt | 3h21m | 9m | 15m | 1h7m | 21h | 1h24m | 1h16m | 10m |
>
> #### **Q2: Should compare with LoRA**
>
> **A2:** Thank you for the suggestion. We agree with the reviewer that LoRA is an important parameter-efficient tuning method. We conduct additional experiments and report the comparison results (using T5-XL) below. We will incorporate these results in Table 7 during revision.
>
> |  Method  | Boolq | CB | COPA | MRC | ReC | RTE | WiC | WSC |
> | :-: | :-: | :-: | :-: | :-: | :-: | :-: | :-: | :-: |
> | Fine-Tuning | 89.60 | 94.20 | 96.00 | 76.15 | 92.05 | 91.70 | 74.30 | 95.20 |
> | **LoRA** | 89.48 | 94.66 | 96.15 | 75.78 | 92.36 | 91.91 | 74.49 | 95.82 |
> | APrompt | 90.72 | 95.48 | 95.83 | 78.68 | 93.75 | 93.36 | 76.43 | 96.17 |
>
> #### **Q3: Missing references**
>
> **A3:** Thank you for pointing these related works to us! We will add references [2] and [3] in the revised paper ([1] is already cited).
>
> Thanks again for your review! Please let us know if you have any further questions, and we are happy to discuss further.

---

### Official Review · Reviewer_97Ph · 2023-08-03

**Soundness:** 4

**Excitement:**

3: Ambivalent: It has merits (e.g., it reports state-of-the-art results, the idea is nice), but there are key weaknesses (e.g., it describes incremental work), and it can significantly benefit from another round of revision. However, I won't object to accepting it if my co-reviewers champion it.

**Paper Topic And Main Contributions:**

This paper proposes another parameter efficient fine-tuning method of large language models: attention prompt tuning. APROMPT incorporates three sets of learnable prompts: query, key, and value prompts. These prompts are prepended to the respective matrices in the self-attention block within the Transformer layer. During model tuning, these attention prompts are learned alongside the original input prompts. Comprehensive experiments on various tasks in the SuperGLUE benchmark show the effectiveness of the proposed method.

**Reasons To Accept:**

1. Clear writing.
2. Thorough analysis and discussion.

**Reasons To Reject:**

My main concern of the paper is on the baselines. As newly-proposed parameter efficient fine-tuning method, it should not only be compared with prompt tuning approaches, but also other methods such as Adaptor (https://arxiv.org/pdf/1902.00751.pdf) and LoRA (https://arxiv.org/abs/2106.09685), since they are more widely used.

**Reproducibility:**

4: Could mostly reproduce the results, but there may be some variation because of sample variance or minor variations in their interpretation of the protocol or method.

**Reviewer Confidence:**

4: Quite sure. I tried to check the important points carefully. It's unlikely, though conceivable, that I missed something that should affect my ratings.

---

> ### Author Rebuttal · Authors · 2023-08-27
>
> We sincerely thank you for your thoughtful feedback!
>
> #### **Q1: Compare with non-prompt tuning approaches**
>
> **A1:** Thank you for your great suggestion! We completely agree with the reviewer that it is important to compare with other non-prompt tuning based parameter efficient approaches. In fact, we have compared with several widely used parameter efficient methods and reported the results in Appendix C (Table 7). The Adaptor (we should have named it Adaptor-H) method in Table 7 is a v2 version of the Adaptor work the reviewer mentioned. For completeness, we report the comparison results with the two important methods pointed out by the reviewer below (using T5-XL). We’ll include these additional results in our revised paper. Thank you again for the suggestion.
>
> |  Method  | Boolq | CB | RTE | WSC |
> | :-: | :-: | :-: | :-: | :-: |
> | Fine-Tuning | 89.60 | 94.20 | 91.70 | 95.20 |
> | **Adapter** | 84.46 | 85.89 | 85.53 | 88.75 |
> | Adapter-H | 85.37 | 86.54 | 85.75 | 89.17 |
> | **LoRA** | 89.48 | 94.66 | 91.91 | 95.82 |
> | APrompt | 90.72 | 95.48 | 93.36 | 96.17 |
>
> Thanks again for your review! Please let us know if you have any further questions, and we are happy to discuss further.

---

### Official Review · Reviewer_tGWP · 2023-08-05

**Typos Grammar Style And Presentation Improvements:** In Figure 3 (c), Matmul -> MatMul
**Soundness:** 4

**Excitement:**

3: Ambivalent: It has merits (e.g., it reports state-of-the-art results, the idea is nice), but there are key weaknesses (e.g., it describes incremental work), and it can significantly benefit from another round of revision. However, I won't object to accepting it if my co-reviewers champion it.

**Paper Topic And Main Contributions:**

The paper presents Attention Prompt tuning (APrompt), an innovative method for adapting large pre-trained language models. Unlike traditional tuning that focuses on input layers, APrompt incorporates prompts into the attention layer. Tested on the SuperGLUE benchmark, APrompt outperforms standard approaches across different model scales, with additional studies confirming its effectiveness and efficiency.

**Questions For The Authors:**

Please refer to [reasons to reject].

**Reasons To Accept:**

1. Views the existing prompt tuning method as a specialized form of attention prompts.

2. Proposes a novel prompting method where the prompts are applied to the attention directly.

3. Extensive experiments on three different model size and SuperGLUE benchmark shows the effectiveness of the proposed method.

4. The paper is well-written and the figures help the understanding.

**Reasons To Reject:**

1. I have a hard time understanding why in line 213, Q is not calculated using $X_{new}$. It seems like the authors argue this because only the original text tokens X are updated. However, to the best of my knowledge, the features of the prompts are also updated as it goes through the transformer layers.

2. In Table 8, the authors report the training time of APrompt on the SuperGLUE dataset. But I feel its necessary to report the training time of other baselines as well.

**Reproducibility:**

4: Could mostly reproduce the results, but there may be some variation because of sample variance or minor variations in their interpretation of the protocol or method.

**Reviewer Confidence:**

5: Positive that my evaluation is correct. I read the paper very carefully and I am very familiar with related work.

---

> ### Author Rebuttal · Authors · 2023-08-27
>
> We sincerely thank you for your thoughtful feedback! We discuss your raised points as follows:
>
> #### **Q1: Regarding Q is not calculated using X_new**
>
> **A1:** Thank you for your insightful question. We'd like to provide some clarification here. In the original Prompt Tuning work (Lester et al., 2021), the prompts are exclusively applied or appended to the input layer of the transformer, as illustrated in the second subfigure of Figure 1. It's important to note that while it's possible to compute updated features for these prompts after the first transformer layer, these prompts are not utilized in subsequent layers of the model.
>
> In both P-Tuning V2 (Liu et al., 2022) and our own approach, the prompts are prepended to all layers of the transformer, as depicted in the two rightmost subfigures of Figure 1. However, it's crucial to emphasize that the prompts from different layers are learned independently and are not updated based on information from their previous layers. To illustrate this further, if you closely examine Figure 3(a), you'll observe that the prompts in the next layer are not updated or calculated from the previous layer (no connections to the previous layer). Meanwhile, the features of the text sequence are indeed updated from the previous layer.
>
> In summary, in prompt tuning approaches, only the original text tokens are updated through the transformer layers. We thank you again for pointing this out. We will incorporate the above discussion during revision to make it more clear.
>
> #### **Q2: Regarding training time of other baselines**
>
> **A2:** Thank you for the great suggestion! We report the training time of all methods below (training time of Fine-Tuning is not reported as we directly import the results from the original paper). We’ll supplement these results into our revision.
>
> |  Method  | Boolq | CB | COPA | MRC | ReC | RTE | WiC | WSC |
> | :-: | :-: | :-: | :-: | :-: | :-: | :-: | :-: | :-: |
> | Prompt-Tuning| 2h38m | 8m | 11m | 46m | 18h25m | 57m | 51m | 8m |
> | P-Tuning v2| 3h36m | 10m | 14m | 1h15m | 20h12m | 1h28m | 1h14m | 12m |
> | XPrompt| 4h47m | 14m | 29m | 1h53m | 28h41m | 2h26m | 2h19m | 19m |
> | ResPrompt| 3h47m | 10m | 23m | 1h21m | 22h32m | 1h51m | 1h34m | 12m |
> | APrompt | 3h21m | 9m | 15m | 1h7m | 21h | 1h24m | 1h16m | 10m |
>
> #### **Q3: Regarding the typo**
>
> **A3:** Thank you for pointing it out. We will revise accordingly.
>
> Thanks again for your review! Please let us know if you have any further questions, and we are happy to discuss further.

---

### Meta-Review · Area_Chair_AJvZ · 2023-09-07

**Recommendation:** 4

**Metareview:**

The paper introduces APrompt as a parameter-efficient fine-tuning method for large PLMs. Rather than modifying input layers, APrompt integrates query, key, and value prompts into the attention layer of the transformer. These prompts are learned alongside the original input prompts during model tuning. APrompt's effectiveness is demonstrated through comprehensive experiments on various SuperGLUE benchmark tasks, outperforming standard approaches across different model scales. Additionally, the study highlights that existing prompt tuning can be seen as a special case of attention prompt tuning, emphasizing the method's practicality and efficiency.

All reviewers engaged in extensive discussions with the authors and all agree the contribution of this work is strong. All reviewers agree that the results are reproducible and the excitement of this work is more than average.

---

### Decision · Program_Chairs · 2023-10-07

**Decision:**

Accept-Main

**Comment:**

The paper introduces APrompt as a parameter-efficient fine-tuning method for large PLMs. Rather than modifying input layers, APrompt integrates query, key, and value prompts into the attention layer of the transformer. These prompts are learned alongside the original input prompts during model tuning. APrompt's effectiveness is demonstrated through comprehensive experiments on various SuperGLUE benchmark tasks, outperforming standard approaches across different model scales. Additionally, the study highlights that existing prompt tuning can be seen as a special case of attention prompt tuning, emphasizing the method's practicality and efficiency.

All reviewers engaged in extensive discussions with the authors and all agree the contribution of this work is strong. All reviewers agree that the results are reproducible and the excitement of this work is more than average.